

# The global impact of the transport sectors on aerosol and climate under the Shared Socioeconomic Pathways (SSPs)

Mattia Righi, Johannes Hendricks, and Sabine Brinkop

Deutsches Zentrum für Luft- und Raumfahrt (DLR), Institut für Physik der Atmosphäre, Oberpfaffenhofen, Germany

**Correspondence:** Mattia Righi (mattia.righi@dlr.de)

**Abstract.** A global aerosol-climate model is applied to quantify the impact of the transport sectors (land transport, shipping and aviation) on aerosol and climate. Global simulations are performed for present-day (2015), based on the emission inventory of the Climate Model Intercomparison Project Phase 6 (CMIP6), and for near-term (2030) and mid-term (2050) future projections, under the Shared Socioeconomic Pathways (SSPs). The results for present-day show that land transport emissions have a large impact on near-surface concentrations of black carbon and aerosol nitrate over the most populated areas of the globe, but with contrasting patterns in terms of relative contributions between developed and developing countries. In spite of the recently introduced regulations to limit the fuel sulphur content in the shipping sector, shipping emissions are still responsible for a considerable impact on aerosol sulfate near-surface concentrations, about 0.5 to 1 $\mu$g m$^{-3}$ in the most travelled regions, with significant effects also on continental air pollution and in the northern polar regions. Aviation impacts on aerosol mass are found to be quite small, of the order of a few nanograms per cubic meter, while this sector considerably affects particle number concentrations, contributing up to 20–30% of the upper tropospheric particle number concentration at the northern mid-latitudes. The transport-induced impacts on aerosol mass and number concentrations result in a present-day radiative forcing of $-164$, $-145$ and $-64$ mW m$^{-2}$, for land transport, shipping and aviation, respectively, with a dominating contribution by aerosol-cloud interactions. These forcings represent a marked offset to the CO$_2$ warming from the transport sectors and are therefore very relevant for climate policy. The projections under the SSPs show that the impact of the transport sectors on aerosol and climate are generally consistent with the narratives underlying these scenarios: the lowest impacts of transport on both aerosol and climate are simulated under SSP1, especially for the land transport sector, while SSP3 is generally characterized by the largest effects. Notable exceptions to this picture, however, exist, as the emissions of other anthropogenic sectors also contribute to the overall aerosol concentrations and thus modulate the relevance of the transport sectors in the different scenarios, not always consistently with their underlying storyline. On a qualitative level, the results for present-day mostly confirm the findings of our previous assessment for the year 2000, which used a predecessor version of the same model and the CMIP5 emissions data. Some important quantitative differences are found, which can mostly be ascribed to the improved representation of aerosol background concentrations in the present study.





## 1 Introduction

With a share of 14% (8.5 Pg) of the total $CO_2$-equivalent emissions in 2018, the transport sectors are major contributors to climate change (Lamb et al., 2021). This share is dominated by land transport (including road and rail transport, and inland navigation), which is responsible for 80.4% of it, while international shipping and aviation (international and domestic) account for the remaining 8.4% and 11.2%, respectively. Moreover, the transport sectors are characterized by an approximately steady growth in emissions of greenhouse gases, at a rate of about 2% per year in the last three decades, mostly driven by developed

countries (North America and Europe) and, since the beginning of the century, by the fast growing economies of East Asia (Lamb et al., 2021). Traffic volumes are projected to grow further in the coming decades (Girod et al., 2013; Sims et al., 2014), modulated by the current efforts to decarbonise this sector in order to achieve the goals of the Paris Agreement (Pietzcker et al., 2014; Creutzig et al., 2015; Esmeijer et al., 2020) and by the uncertain recovery path from the COVID-19 pandemic, which significantly affected the recent global emissions (Liu et al., 2020; Le Quéré et al., 2020).

Although the climate policies mostly target the emissions of $CO_2$ and other greenhouse gases, the combustion processes powering vehicles, ships and aircraft are also responsible for the emissions of several types of short-lived pollutants, such as nitrogen oxides ($NO_x=NO+NO_2$), carbon monoxide (CO), non-methane hydrocarbons (NMHC), sulphur dioxide ($SO_2$) and aerosol particles in form of black carbon (BC), organic carbon (OC) and primary aerosol sulfate ($SO_4$). These compounds have a detrimental effect on the air quality, while some of them can also impact climate (Fiore et al., 2012; von Schneidemesser

et al., 2015). For example, $NO_x$ and NMHC are precursors of ozone, which is a well-known greenhouse gas (e.g. Stevenson et al., 2013; Mertens et al., 2018), while aerosol and their precursors can alter the planetary radiation balance via their direct interaction with solar radiation (absorption and scattering, leading to warming and cooling, respectively; Boucher et al., 2013). Perhaps most importantly, aerosol can further impact the climate by interacting with clouds and modifying their radiative properties, resulting in an overall cooling effect (Bellouin et al., 2020). Air pollution reduction measures targeting these compounds

may therefore affect both air quality and climate, with beneficial effects for both in some cases (e.g. when targeting BC, which has a warming effect on climate and a detrimental impact on air quality; Bond et al., 2013), and trade-offs in other cases, like sulphur reduction policies, which improve air quality but reduce the cooling impact due the aerosol indirect effect on clouds (Fuglestvedt et al., 2009). Furthermore, these policies are implemented at different scales, with land transport being mostly targeted at the national or regional (e.g. European) level, while shipping and aviation are regulated at the international level,

via the International Maritime Organization (IMO) and the International Civil Aviation Organization (ICAO), respectively, although regional policies are also applied in shipping, for instance the implementation of the Sulphur Emissions Control Areas (SECAs; Buhaug et al., 2009; Smith et al., 2014). To fully characterize the impact of short-lived pollutants from the transport sectors it is therefore necessary to consistently account for changes in the atmospheric composition and climate impact and, at the same time, to address regional dependencies.

In this study, we focus on the impact of transport-induced aerosol and aerosol precursor species. We update the assessments of Righi et al. (2013, 2015, 2016) and improve them on three major aspects:





1. We use the recent CMIP6 emission inventory (Hoesly et al., 2018; van Marle et al., 2017), thus replacing the CMIP5 inventory (Lamarque et al., 2010) applied in our previous studies and now consider the year 2015 instead of 2000 as a reference for present-day conditions.

2. We implement the Shared Socioeconomic Pathways (SSPs; O'Neill et al., 2017) for the future projections, which are a new generation of scenarios developed in support of the IPCC Sixth Assessment Report (Gidden et al., 2019). They are socio-economic scenarios that include assumptions on several indicators (e.g. population, wealth, urbanisation, etc.) to describe the ability of the society to cope with climate change and are further combined with the climate pathways based on the narratives of Representative Concentration Pathways (RCPs; van Vuuren et al., 2011) for climate policies, resulting in a matrix of possible futures.

3. We apply a newer version of the global chemistry-climate model EMAC (ECHAM-MESSy Atmospheric Chemistry; Jöckel et al., 2010), featuring an updated aerosol scheme (MADE3, i.e. 3rd version of the Modal Aerosol Dynamics for Europe adapted for global applications; Kaiser et al., 2014) and a state-of-the-art two-moment cloud scheme (Kuebbeler et al., 2014) accounting for aerosol-cloud interactions for all cloud phases (liquid, mixed-phase and ice). This model configuration has been thoroughly tuned and evaluated by Kaiser et al. (2019) and Righi et al. (2020), demonstrating its ability to represent key aspects of aerosol, cloud and radiation processes.

Being key targets of decarbonisation and air pollution control policies, the transport sectors are subject of active research. Lund et al. (2014a) applied a global chemistry-transport model to quantify the effect of short-lived pollutants on climate when replacing conventional diesel fuels with biofuels in the EU road transport sector and found a 15% and 80% reduction of radiative forcing (RF) when replacing 20% and 100% of diesel with biofuels, respectively. Lund et al. (2014b) considered the climate impact of black carbon from road diesel engines in 2010 and 2050 following a current-legislation scenario and found that Europe gives the largest contribution in 2010, while South and East Asia dominate in 2050, as a result of different implementation of reduction measures. The potential global relevance of local policies on land-based transport was highlighted by Hendricks et al. (2018), who estimated the climate impact of German land-based transport using a combination of modelling techniques and found that this is dominated by $CO_2$, while the effect of short-lived pollutants is negligible. They also note, however, that this might be different in other regions or countries, depending on the specific composition of the fleet and on its age, and also on the photochemical activity which varies considerably around the globe. The climate impact of ozone from global land-transport and shipping was quantified by Mertens et al. (2018), who applied an innovative tagging method to track the emissions from specific sectors in a global model simulation and thus isolate the climate impact with better accuracy than in standard perturbation approaches. This resulted in higher land-transport- and shipping-induced concentrations of surface-level ozone and in a RF of 92 and 62 $\mathrm{mW\,m^{-2}}$, for the two sectors respectively. The shipping sector is particularly interesting due to the new regulations introduced by MARPOL 73/78 (the International Convention for the Prevention of Pollution from Ships) in its Annex VI to globally limit fuel sulphur content (FSC) in shipping (Buhaug et al., 2009) and which took full effect in 2020. Sofiev et al. (2018), Kontovas (2020) and Bilsback et al. (2020) showed how reducing FSC leads to reduced $SO_2$ emissions and lower shipping-induced aerosol concentrations: while this is beneficial for air quality and reduces premature mortality, it may





also remove the cooling from the aerosol effects, hence accelerating global warming. Measurements of ship engine exhaust, however, show that particle number emission factors in low-sulphur fuels are not significantly decreased compared to standard heavy fuel oil (Kuittinen et al., 2021), implying that shipping would remain an important source of particle number even after the MARPOL regulations. Another policy-relevant issue is the opening of Arctic routes in the near future (Stephenson et al.,
2018) and its impacts on the regional and global climate.

As shown in Righi et al. (2013, hereafter R13), FSC is also critical for the aviation sector, as sulfate particles produced in the aircraft exhaust plume could be transported downwards and potentially impact low level clouds, resulting in a significant (albeit quite uncertain) aerosol indirect effect. R13 estimated this effect to be between $-70$ and $-15\,\mathrm{mW\,m^{-2}}$, depending on the assumptions on the size of the emitted sulphate particles, but found no statistically significant effect when a low FSC was
assumed for the aircraft fleet. This effect was also simulated by Gettelman and Chen (2013), who however considered generally smaller particle sizes for aerosol sulfate in their assumptions, thus finding a larger RF in the range $-164$ to $-23\,\mathrm{mW\,m^{-2}}$. A later study by Kapadia et al. (2016) supported these estimates and again found a reduction of the indirect effect under a scenario with low aviation FSC, while Matthes et al. (2021) showed that the impact of aviation aerosol on low clouds may be dependent on the cruise altitude of the global fleet. Aviation emissions of soot are also potentially impacting ice clouds
(Hendricks et al., 2005, 2011; Penner et al., 2018), but this effect is very uncertain and strongly dependent on the assumed ice nucleating abilities of soot. In the model configuration adopted in the present study, we conservatively assume a very low ice nucleating efficiency for aviation soot, so that its impact on climate via the interaction with natural cirrus clouds may be considered negligible (Righi et al., 2021).

Despite active research on the impact of the transport sectors on atmosphere and climate, the literature on the impact of
specific sectors in the context of the SSPs is still limited and a detailed analysis on the role of transport-induced aerosol is still lacking. Lund et al. (2019) assessed the anthropogenic aerosol forcing in a subset of the SSP scenarios until the end of the century, concluding that the air pollution control measures assumed in these scenarios span a much broader range of emissions than in the previous RCPs. Lund et al. (2020) used a climate response emulator to calculate the temperature response of various short-lived compounds under the SSPs and compared it to the one of $CO_2$, showing that the role of short-lived compounds on
climate is projected to remain relevant throughout the century, with strong regional variations. Hence, re-assessing the impact of transport on aerosol concentrations, also with a focus on regional effects, is essential to understand the scope of the air pollution measures covered by the SSPs narratives and the resulting climate impacts.

This paper is organised as follows: Section 2 briefly describes the EMAC model with the aerosol submodel MADE3 and the method used to estimate the impact of the transport sectors. A detailed analysis of the emission inventories for present-day
and the future projections is provided in Sect. 3, focusing on the share of the three transport sectors on the total emissions of aerosol and precursors. The impact of transport on the aerosol concentration is discussed in Sect. 4, while Section 5 deals with the climate impacts. A summary of the conclusions of this study is provided in Sect. 6.



## 2 Model and model simulations

In this assessment we apply the global model EMAC with the aerosol submodel MADE3. EMAC is a chemistry and cli-
mate model featuring several submodels to simulate tropospheric and middle atmospheric processes and their interaction with
oceans, land, and human influences (Jöckel et al., 2010). EMAC is based on the second version of the Modular Earth Submodel
System (MESSy) to link multi-institutional computer codes. The core atmospheric model is the ECHAM5 (fifth generation Eu-
ropean Centre Hamburg) general circulation model (Roeckner et al., 2006). Aerosols microphysics is represented by the aerosol
submodel MADE3 (Kaiser et al., 2014), which considers aerosol sulfate, ammonium, nitrate, sea salt, particulate organic mat-
ter, black carbon, mineral dust, and aerosol water. These species are distributed in nine aerosol modes, resulting from the
combination of three size classes (Aitken, accumulation, and coarse) with three particle mixing states (soluble, insoluble and
mixed particles). The representation of aerosol mass and number concentrations in MADE3 has been evaluated in detail in
Kaiser et al. (2019), using observational data from various station networks around the globe, from aircraft campaigns, and
from satellite observations.

For the present study, we apply the model configuration developed by Righi et al. (2020, 2021), which further includes an
explicit representation of aerosol-radiation and aerosol-cloud interactions. Cloud microphysics and precipitation are dealt with
by the CLOUD submodel, which in this configuration considers aerosol effects on warm, mixed-phase and cirrus clouds based
on the two-moment cloud scheme by Kuebbeler et al. (2014). Cloud-radiation and aerosol-radiation interactions are simulated
by the submodels CLOUDOPT and RAD (Dietmüller et al., 2016), respectively. As the focus of this study lies on the radiative
effects of aerosol and clouds, the concentrations of the radiatively active gases $CO_2$, $CH_4$, $N_2O$, $O_3$ and chlorofluorocarbons
are prescribed using globally averaged values for the year 2015. The studies by Kaiser et al. (2019) and Righi et al. (2020)
demonstrated the model's ability to reproduce the relevant observed aerosol properties both close to the surface and in the
middle and upper troposphere, as well as the main cloud and radiative properties. This configuration of EMAC is therefore
suited to the scope of the present study.

The chosen model resolution is T42L41, corresponding to a horizontal resolution of about $2.8^{\circ} \times 2.8^{\circ}$ in latitude and longi-
tude (roughly 300 km at the Equator), with 41 non-equidistant vertical levels from the ground to about 5 hPa, mostly centered
within the troposphere. All simulations cover a period of 11 years, between 2005 and 2015, with the first year used as spin-up
and hence not included in the analysis. The model meteorology (temperature, winds and logarithm of the surface pressure) is
nudged using the ERA-Interim reanalysis data for the same time period (Dee et al., 2011).

As in R13, the effects of the transport sectors are quantified using the perturbation approach, in which a reference simulation
is compared to a sensitivity one where the emissions of a given sector are completely neglected. In the following sections, we
show both the absolute change ($A$) due to the emissions from the given sector and their relative contribution ($R$) to a specific
quantity (e.g., burden or concentration). Considering, for instance, the land transport sector, these are defined as:

$$A = \mathrm{REF} - \mathrm{NOLAND} \tag{1}$$




$$R = (\text{REF} - \text{NOLAND})/\text{REF} \tag{2}$$

where REF is the reference simulation and NOLAND the sensitivity simulation with land transport emissions switched off. The same applies to the other two sectors, i.e. shipping and aviation. The statistical significance of $A$ and $R$ is verified by means of a two-tailed t-test with respect to the interannual variability. We consider the results to be statistically significant when their
confidence level is above 95%.

As shown by R13, the perturbation method outlined above could lead to some inaccuracies in the estimated effects, due to the non-linearities in the involved processes, which affect in particular the secondary aerosol species. Alternative methods, such as for example tagging (Grewe, 2013; Rieger et al., 2018; Mertens et al., 2018), are hard to apply to the aerosol phase, especially for secondary species, due to the difficulties in tracking the chemical processes taking place both in the gas phase
and in the liquid phase (e.g., aerosol sulfate formation). Furthermore, the bulk of the climate effects estimated here are expected to result from aerosol interactions with clouds, which cannot be isolated with the tagging approach, due to the prominent role of feedbacks.

## 3 Emissions of aerosol and precursors from the transport sectors

The model simulations performed in this study are driven by the anthropogenic and open burning emission datasets developed
in support of the CMIP6 Project (Hoesly et al., 2018; van Marle et al., 2017). The emission data cover the historical period from the pre-industrial time (1750) to present-day (2014), and future projections until the year 2100. Here, we consider a present day case (2015) and two projections, for a near- (2030) and mid-term (2050) future, both under three SSPs: SSP1 (van Vuuren et al., 2017), SSP2 (Fricko et al., 2017), and SSP3 (Fujimori et al., 2017). Although more SSPs are available in this inventory, the three SSPs analysed here are fairly representative of possible futures, including: a sustainability scenario (SSP1) with
strong technology development and environmental concerns; a middle-of-the-road scenario (SSP2) reflecting a continuation of historical trends in many aspects; and a regionally heterogeneous scenario (SSP3) characterized by slow economic growth, weak technology development and, most importantly for the present assessment, by a high level of air pollutant emissions, including aerosol. We do not consider the long-term future after 2050, as the projections beyond this time horizon are affected by large uncertainties in their driving forces (such as demographic changes, economic growth and technological development;
see Bauer et al., 2017). The reason for choosing 2015 (from the SSP2 scenario) as the reference for the present day, instead of the last year of the historical CMIP6 time-series (2014), is to have consistent emissions from the same database, i.e. the SSPs. Note, however, that the small differences between 2014 and 2015 are negligible for the scope of the present assessment.

A series of specific assumptions corresponds to each of the SSP narratives, concerning, for instance, population and economic growth, urbanisation level, policy implementation, technological advancements, availability of resources, human and
societal developments, etc. These assumptions are then elaborated by integrated assessment models to produce emission projections for greenhouse gases, short-lived gases and aerosol particles. Note that in their reference version, the SSPs storylines



do not assume any implementation of climate policies. In the matrix framework of the SSPs (van Vuuren et al., 2014), climate policies are included at a later stage, when the reference scenarios are combined with mitigation scenarios using the same approach as in the RCPs (Moss et al., 2010; van Vuuren et al., 2011), i.e., targeting a given anthropogenic RF in the year 2100.

Among the various combinations of the SSPs with forcing pathways (Gidden et al., 2019), we choose here SSP1-1.9 (targeting an anthropogenic RF of 1.9 W m$^{-2}$ in 2100 and matching the 1.5°C goal of the Paris Agreement), SSP2-4.5 (RF of 4.5 W m$^{-2}$ in 2100, close to the previous RCP4.5 scenario), and SSP3-7.0 (RF of 7.0 W m$^{-2}$ in 2100). For simplicity, in the following we will refer to the SSPs just by their number and skip the RF indication, as there is no ambiguity.

The total emissions of aerosol and precursor gases from all sectors in the CMIP6 inventory are summarized in Fig. 1, for
the present-day (2015) and the three SSPs for the years 2030 and 2050. The total emissions of the anthropogenic non-transport sectors (comprising energy, industry, residential and commercial, agriculture, solvents production and application, and waste) dominate the emission budget of all species in all scenarios, with relative contributions over 90% to NH$_3$ emissions (Fig. 1c) mostly due to the agriculture sector. The transport sectors are particularly relevant for NO$_x$ emissions (Fig. 1a), with large relative contributions from land transport and shipping (mainly driven by emissions from diesel engines) and BC (Fig. 1e,
especially for land transport). Shipping and aviation contributions are generally small, but one should note that ships and aircraft mostly travel in regions which are otherwise hardly affected by anthropogenic emissions, hence their local contribution to atmospheric aerosol concentrations can be relevant even though the emissions are low.

Future emissions in 2030 and 2050 strongly decrease with respect to present day in SSP1, consistently with the narrative underlying this scenario. According to van Vuuren et al. (2017), SSP1 is characterized by a rapid decline in costs for clean
technologies, favoring a rapid transition to electric vehicles, and by an increase in public transport and car sharing. It further assumes the worldwide implementation of strict air-quality policies, currently only applied in OECD countries. The high share of electric vehicles (up to 75%) leads to a strong decrease in NO$_x$ emissions in SSP1, whose contribution to the total is reduced from 26% in 2015 to 16% in 2030 and further down to 6.7% in 2050, while the contribution of aviation remains constant around 2% and the shipping one is only reduced in 2050, due to the known challenges in electrifying these sectors. A similar trend can
be seen for the other species. The reduction in SO$_2$ emissions is stronger than for other species in SSP1, mostly driven by the decrease in the non-traffic anthropogenic emissions, which, however, maintain a roughly constant share around 85–90% of the total. The shipping sector contribution to SO$_2$ remains relatively high until 2030, decreasing only slightly from 9% in 2015 to 7% in 2030. This could be due to the phase-out of traditional (low-sulphur) biofuels in 2030 in this scenario and to the lack of alternative technologies before 2050, when the share is strongly reduced to 1.7%.

In contrast to the optimistic SSP1 scenario, the SSP2 narrative (Fricko et al., 2017) assumes a significant use of fossil fuel in the future, combined with less stringent air pollution reduction measures. This leads to an only moderate reduction in the emissions of short-lived compounds in the future. The share of NO$_x$ emissions by land-transport and shipping remains approximately constant until 2050 (around 25% and 14%, respectively), while for aviation it almost doubles in 2030 (from 1.9% to 3.4%) and triples in 2050 (5.3%). This is related to the increase in air traffic volumes combined with the lack of
efficient alternative technologies for emissions reduction in this sector, which only in SSP1 is counteracted by a reduction in air traffic. Similar trends can be seen for the other species in Fig. 1, with the exception of SO$_2$ for shipping. In this case, a very





**Figure 1.** Total emissions of each species from the different sectors in the CMIP6/SSP inventory, compared with the year 2000 emissions from the CMIP5 inventory (Lamarque et al., 2010) used in Righi et al. (2013). The numbers on the left of each bar indicate the relative contribution (in percent) of each sector to the total. OC emissions are multiplied by a factor 1.4 in the model to obtain particulate organic matter (POM) emissions, as required by the MADE3 aerosol submodel. Furthermore, a small fraction ($\sim 2\%$, depending on the sector) of the $SO_2$ emitted mass is emitted as primary aerosol sulfate in the model (see Kaiser et al., 2019, for more details). Note that the vertical scales differ among the panels.





marked decrease in the share of this sector is evident in SSP2, decreasing from 9.1% in 2015 to 3.1% in 2030 and 0.4% in 2050, representing the lowest share of this sector among the three investigated scenarios.

SSP3 represents a more pessimistic scenario, implementing weak air pollution reduction measures, combined with continued
dependency on fossil fuels and slow implementation of new technologies (Fujimori et al., 2017). This scenario further considers a late saturation in transport demand, combined with a low electrification level (up to 10%). This leads to an increase in the land-transport share of $NO_x$ which, together with the increase in the other non-traffic anthropogenic sources, drives the overall increase of $NO_x$ emissions in SSP3 in 2030. The total $NO_x$ emissions remain then approximately constant until 2050, thanks to the reduced share of the shipping sector and the shift to alternative fuels. Due to the rapid shift to high speed transport modes
assumed in SSP3, the share of aviation increases as well, from 1.9% in 2015 to 2.6% in 2030 and 3.3% in 2050, although this increase is less marked than in SSP2. An analogous trend is also evident for $SO_2$, BC and OC emissions, although the latter is partly compensated by a reduction of open burning emissions in 2030 and 2050.

Since in the present work we use an updated inventory (CMIP6/SSP) with respect to our previous assessment (CMIP5/RCP; R13), in Fig. 1 we also compare the total emissions between these two inventories. Note that not only the inventory was
updated, but we also considered a different baseline year for present-day conditions (2015 instead of 2000). A general increase in the total emissions can be seen, mostly driven by the anthropogenic non-transport sectors, while only the emissions of CO (Fig. 1b) and $SO_2$ (Fig. 1d) slightly decrease. The transport sectors mostly show a decrease in emissions between 2000 and 2015, both in absolute and relative terms, with the notable exception of $NO_x$ (Fig. 1a) and BC (Fig. 1e), whose share, however, decreased from 29% to 26% and from 17% to 14%, respectively. This indicates that, although the overall growth
of land transport volumes leads of course to an increase in emissions of these two highly relevant species for this sector, the introduction of air pollution control measures (e.g., on diesel engines) in the developed countries helped to reduce the relative impact of the sector on the global emission budget. Another interesting case is $SO_2$ (Fig. 1d) from shipping, which shows a decrease in both absolute and relative terms between 2000 and 2015. This is a consequence of the limits on shipping FSC introduced starting from 2010 in the MARPOL Annex VI (Buhaug et al., 2009). These measures also affected shipping $NO_x$
emissions, which however still grew between 2000 and 2015 (Fig. 1a), but keeping a stable share of 14% of the total. The emissions from the aviation sector increase for all species between 2000 and 2015, which is expected since emission reduction measures in this sector are over-compensated by the high growth rates in traffic volumes during this period. An important improvement in the CMIP6 inventory is the consideration of the emissions from more species in the aviation sector, while in CMIP5 only $NO_x$ and BC were available. In R13 we nevertheless included $SO_2$ emissions from aviation, deriving them from
BC emissions by scaling with the respective emission factors of the two species.

The emission inventories applied here only provide data for the emitted mass of each compounds, whereas the MADE3 aerosol submodel requires emission fluxes for both mass and number. The latter is derived by assuming typical log-normal size distributions for the emitted particles in each sector, based on measurements when available. Further assumptions are made to assign the emitted mass to the various modes of MADE3. This follows the same approach of Kaiser et al. (2019), summarized
in Table 1. For the open burning sector, we follow the AeroCom recommendations (Dentener et al., 2006) and assign the aerosol emissions of primary sulfate ($SO_4$), BC and OC to the accumulation mode. For the anthropogenic non-traffic sector, we use the





**Table 1.** Summary of the emission setup used for the model simulations in this study, together with the log-normal size distribution parameters applied to derive number emission fluxes from the mass. MF is the mass fraction in the Aitken (akn) and accumulation (acc) mode. $D$ (geometric mean diameter) and $\sigma$ (standard deviation) are the parameters of the log-normal distribution. See Kaiser et al. (2019) and references therein for a detailed discussions about these parameters.

| Sector | Specie | $MF_{akn}$ [%] | $D_{akn}$ [nm] | $\sigma_{akn}$ | $MF_{acc}$ [%] | $D_{acc}$ [nm] | $\sigma_{acc}$ |
|---|---|---|---|---|---|---|---|
| Open burning | $SO_4$ | – | – | – | 100 | 80 | 1.8 |
| | BC/POM | – | – | – | 100 | 80 | 1.8 |
| Antrop. non-traffic | $SO_4$ | – | – | – | – | – | – |
| | BC/POM | – | – | – | 100 | 138 | 1.59 |
| Land transport | $SO_4$ | 100 | 30 | 1.80 | – | – | – |
| | BC/POM | 10 | 58 | 1.58 | 90 | 138 | 1.59 |
| Shipping | $SO_4$ | 10 | 70 | 1.45 | 90 | 260 | 1.25 |
| | BC/POM | 10 | 70 | 1.45 | 90 | 260 | 1.25 |
| Aviation | $SO_4$ | 91 | 25 | 1.55 | 9 | 150 | 1.65 |
| | BC/POM | 91 | 25 | 1.55 | 9 | 150 | 1.65 |

parameters measured by Birmili et al. (2009) in the urban background of Berlin, hence assuming an aged aerosol population. This assigns the emissions to the accumulation mode only. Note that no primary $SO_4$ emissions are considered for this sector. We use the same parameters also for land transport, but in this case we assign 10% of the emitted mass to the Aitken mode,

representative of a fresh aerosol population in the urban areas close to the emission sources. Primary $SO_4$ emissions from land transport follow again the AeroCom recommendations, assigning the emitted mass to Aitken mode only (note, however, that the land transport emissions of $SO_4$ are relatively low due to the low-sulphur content of gasoline and diesel fuels used for land transport vehicles in most regions of the world). For shipping, we used the parameters measured by Petzold et al. (2008) in ship plumes. As for the land transport sector, this assumes a 10/90% split of emitted mass between Aitken and accumulation mode,

respectively, but with much larger particles in the accumulation mode (260 nm vs. 138 nm). As noted by Petzold et al. (2008) and R13, due to a more efficient mixing, the aging process is particularly effective in the marine boundary layer, leading to a relatively fast growth of particles in ship plumes. In-situ measurements in actual plumes are also used to derive the parameters for the aviation emissions (Petzold et al., 1999). Due to the high burning efficiency of aircraft turbines, the emitted BC and OC particles are small, with an Aitken mode at 25 nm median diameter comprising about 90% of the emitted mass. Here we

conservatively assume the same size distribution also for primary $SO_4$, although their actual size is much more uncertain and some studies (e.g. Kärcher et al., 2007) show that small, nucleation-size sulfate particles of the order of a few nanometers are present in aircraft plumes during the dispersion phase.





# 4 Transport impact on aerosol concentrations

Fig. 2 shows the impact of land transport, shipping and aviation on the burdens of various aerosol species and of aerosol
particle number. To obtain more representative values, the burdens are calculated integrating the aerosol concentrations over
the domains relevant to the respective sector. For land transport and shipping, the two lowermost model layers (from the surface
to an altitude of about 250 m) are considered, over the continents and over the oceans, respectively. Aviation concentrations are
integrated globally at the typical cruise altitudes between about 8 and 12 km. This approach is consistent with the one adopted
in R13, thus allowing for a direct comparison of the results. In the following, we analyse the impact of the emissions from
the three sectors on the burdens and concentrations of relevant aerosol species, for the present-day (2015) and the three SSP
scenarios in 2030 and 2050. We also compare with the results for the year 2000 obtained with the CMIP5 inventory assessed
by R13. For a correct interpretation of the concentration values discussed in the following, it should be noted that these are
large-scale mean concentrations, which are smaller than the peak concentrations occurring close to the emission sources (e.g.,
at kerbside). Such peak values are highly relevant for assessing air pollution effects, but cannot be resolved by large-scale
climate models which rely on mean concentrations driving large-scale climate effects.

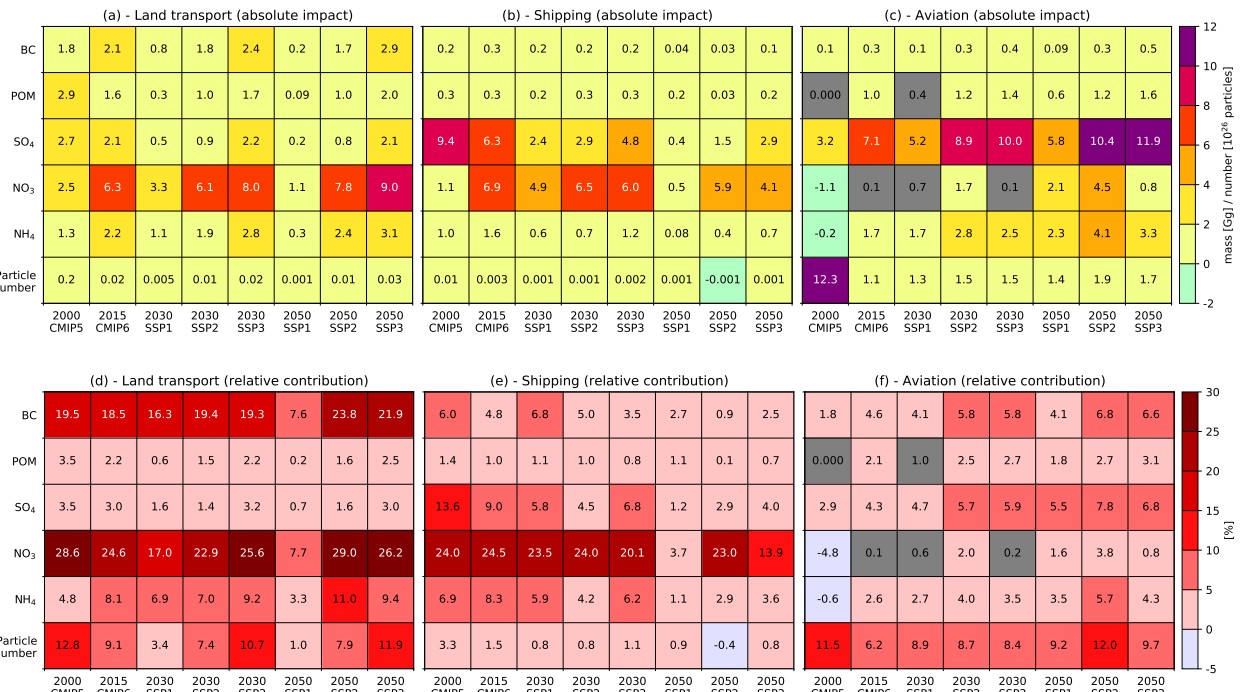

**Figure 2.** Absolute impact (a-c) and relative contribution (d-f) of the three transport sectors to the burdens of various aerosol species and of
particle number. Changes which are not statistically significant at a 95% confidence level are shown in gray. Note the different units for mass
(Gg) and number burdens ($10^{26}$ particles) in panels (a-c).



## 4.1 Land transport

The contribution of land transport to present-day aerosol burdens is particularly large for BC and aerosol nitrate ($NO_3$), see Fig. 2a. The continental near-surface burden of BC is 2.1 Gg in 2015, contributing 19% of the total, similar to the value for the year 2000. The nitrate burden from land transport, on the other hand, is larger in 2015 than in R13 (6.3 vs. 2.5 Gg), but
its relative contribution is slightly lower (25% vs. 29%). This is consistent with the difference in $NO_x$ (a nitrate precursor) emissions between CMIP5 and CMIP6 (Fig. 1a).

In the future scenarios, the absolute impact of land transport on BC is strongly reduced in SSP1 and SSP2 in 2030 and further in 2050, but only in SSP1 this corresponds to a reduction in the relative contributions. On the one hand, this shows that the optimistic SSP1 scenario implements stringent air quality control policies that can effectively reduce land-transport-
induced near-surface concentrations of BC. On the other hand, the higher degree of electrification of this sector in SSP1 (van Vuuren et al., 2017, up to 75%) compared to SSP2 (Fricko et al., 2017, up to 50%) implies strong reductions of BC from land transport, but not from other sectors, hence decreasing the relative contribution of land transport itself. Similar conclusions hold for nitrate, where again only SSP1 is successful in decreasing the land-transport-induced burden both in absolute and relative term, with a very large decrease in 2050.

Transport-induced changes in particle number burden are relevant as well, not only for air quality, but also for climate, since aerosol particles can act as cloud condensation nuclei in warm clouds if they grow to a sufficient size. An increase in their concentration can therefore lead to a change in the microphysical and radiative properties of these clouds (Boucher et al., 2013). Land transport provides a significant contribution to particle number, with 9% of near-surface particle number burden over the continents attributable to this sector in 2015. This is reduced in SSP1 and SSP2 in 2030 and 2050, while it keeps increasing
in SSP3. Note that the impacts on land-transport-induced particle number concentrations is strongly related to the assumptions on the size distribution of emitted particles. As shown in R13, assuming size distribution parameters representative of either young or aged particle populations upon emissions can change the resulting impacts on number concentration by about one order of magnitude. As discussed in Sect. 1 and Table 1, here we assume an aged distribution for particles emitted by land transport, representative of air masses on large spatial scales and therefore more consistent with the representation in the coarse
resolution of a global model.

The geographical distribution of land-transport-induced surface-level BC concentration is shown in Fig. 3. In 2015 (Fig. 3b), the largest absolute impacts are found over China, India and the Arabian Peninsula, with concentrations changes of about 1 $\mu g\,m^{-3}$. Large values are also simulated over Eastern Europe and in some spots in South Africa and South America, in correspondence of the largest urban areas. Western Europe and Eastern U.S. are characterized by generally lower land-transport-
induced BC concentrations. This pattern roughly matches the results for 2000 with the CMIP5 inventory (R13), although with larger values, especially in the Middle East and in Eastern and Southern Asia. In relative terms (Fig. 3c), however, the contribution of land transport is confirmed to be relatively small in Eastern and Southern Asia, amounting to only 10 to 20% of the surface-level BC concentration in these countries, due to the prevalence of emissions from other sectors, like energy production or household. A somewhat opposite effect is the large relative contribution over the Arabian Peninsula, with 60–70% of BC



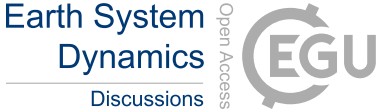

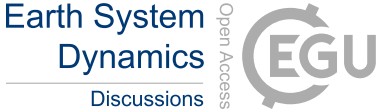

**Figure 3.** Land transport impact on surface-level BC concentration. The left panel show the background concentration for reference. The middle and right panels show the land-transport-induced absolute impacts (Eq. 1) and the relative contributions (Eq. 2), respectively. Grid points where the impacts are not significant to a 95% confidence level are masked out in gray.

**Figure 4.** As in Fig. 3, but for the land transport impact on surface-level particle number concentration.

concentration due to land-transport in these regions, following the comparatively low BC emissions from other sectors in this

region. This sector remains quite relevant for air pollution also in the developed countries, with contributions of 30 to 50% in

Europe and in the U.S. These numbers are lower than in 2000, also in terms of absolute land-transport-induced concentration





changes, revealing the effectiveness of the air pollution control measures and the technical improvements in this sector over the 15 years. Fig. 3c also reveals the importance of the long-range transport of polluted air masses from the continents to the oceans, especially in the subtropics and southern mid-latitudes.

As expected, the land-transport-induced BC concentrations are strongly reduced in SSP1 in 2030 (Fig. 3e) and become almost negligible in 2050 (Fig. 3n), with only India and the Arabian Peninsula showing concentration changes larger than 0.05 $\mu$g m$^{-3}$, although the relative contributions in the Arabian Peninsula still remain high (Fig. 3f,o). This is consistent with the narrative of this scenario, considering a worldwide transition to a more sustainable future with reduced inequality across countries. In SSP2, on the contrary, only the developed countries show a noticeable decrease in BC concentration in 2030 and 2050, both in absolute and relative terms, while developing countries remain on a similar level as in 2015 (Fig. 3h,i). In 2050, land transport is still the major source of BC pollution in many regions of the world, including the Arabian Peninsula, North Africa, and Central and South America (Fig. 3k,l). SSP3 shows no remarkable differences with respect to SSP2 in 2030 (Fig. 3q,r), whereas it is characterized by an increased land-transport-induced concentration in 2050 (Fig. 3t,u), which is again consistent with the storyline of this scenario, assuming large differences between developed and developing countries.

Similar conclusions can be drawn for the other species (Figs. S1–S6), although in relative terms only nitrate (Fig. S1) and particle number concentrations (Fig. 4) show substantial impacts from land transport in 2015 and in the future projections. A remarkable feature is the land-transport-induced decrease in surface-level aerosol sulfate concentration over some parts of India and China (Fig. S3), which could be related to an enhanced aerosol removal via wet deposition. This effect, however, is small in relative terms ($< 10\%$).

With respect to the RCPs scenarios, the projections for the land-transport sectors in the SSPs appear to be more coherent with their narratives in terms of short-lived pollutants, while this was not always the case in the RCPs. Righi et al. (2015), for instance, found that the RCP scenarios with the most stringent climate policies were often characterized by larger impacts on aerosol concentrations, hence showing an anti-correlation between their ranking and the resulting impact on short-lived species. This is not the case in the SSP scenarios: SSP1 is by far the greenest scenario in terms of air pollution impact, while SSP2 and SSP3 show only limited changes in 2030 and 2050 with respect to the present-day situation, consistently with their underlying storylines and long-term climate projection.

### 4.2 Shipping

As in the year 2000, the shipping sector is characterized by its large contributions to the burdens of nitrate and sulfate (SO$_4$, see Fig. 2b) over the oceans. The impact of shipping on near-surface nitrate burden over the oceans in 2015 (6.9 Gg) is significantly larger than in R13 for the year 2000 (1.1 Gg), but it remains stable in relative terms at about 25%. Sulfate pollution, on the other hand, is smaller than in 2000 (6.3 vs. 9.4 Gg) and its relative contribution is reduced to 9% compared to the 14% value reported in R13. As mentioned above, this can be related to the introduction of the MARPOL Annex VI regulations on the maximum FSC in shipping. Although fully implemented in 2020, a first step towards the reduction was taken in 2012, reducing the maximum allowed FSC by mass from 4.5% to 3.5% globally (Buhaug et al., 2009). Stricter limits were applied in the SECAs, reducing the cap from 1.5% to 1% in 2010 and to 0.1% in 2015. This reduction trend appears to continue in the future scenarios,





which are all characterized by a reduced impact of sulfate burden. In 2030, the relative contribution of shipping to near-surface oceanic sulfate burden is reduced to 6–7% in all scenarios, and further to 1–4% in 2050, with the largest (smallest) decrease in SSP1 (SSP3), hence consistent with the scenario narratives. MARPOL also introduced emission limits on $NO_x$ for all new

ship engines built after year 2000 and above a certain power output. These regulations, however, seem to have a limited impact on the resulting nitrate burdens, which are only slightly reduced in 2030 for all scenarios. Stronger reductions in near-surface nitrate burden become effective in 2050, especially in SSP1 (4%) and SSP3 (14%), while they remain high (23%) in SSP2, consistent with the high share of shipping $NO_x$ emissions in this scenario (Fig. 1). The shipping impact on particle number burden is quite small, about 1.5% in 2015, in line with the results of R13. Such a low impact is a consequence of the very

efficient take up of ultra-fine particles by larger aerosol in ship plumes, which occurs within a time scale of the order of only 1 hour (Petzold et al., 2008; Righi et al., 2011). This process is implicitly considered in our assumptions on the size distribution of ship-emitted particles (see Sect. 1 and Table 1), with 90% of shipping emissions released in the accumulation mode, and leads to the small impacts on particle burden. These are further reduced in the future scenarios, in line with the reductions in the mass burden of the most relevant species from this sector.

Fig. 5 depicts the geographical distributions of shipping-induced near-surface sulfate concentrations. In 2015 (Fig. 5b), the largest ship-induced concentrations are simulated over the Northern Hemisphere, especially in the Eastern Atlantic Ocean, in the Northern Indian Ocean and in the Mediterranean, with values around 1 $\mu$g m$^{-3}$. Significant changes of 0.05–0.1 $\mu$g m$^{-3}$ are also simulated over land in many coastal regions of the Northern Hemisphere, confirming the relevance of this sector for continental air pollution (Eyring et al., 2010). This pattern closely matches the distribution simulated in R13 for the year 2000.

The relative contribution of shipping to surface-level sulfate concentration (Fig. 5c), however, is generally lower than in R13. In particular, lower contributions are found over the Northern Pacific Ocean (20–40%, due to the lower absolute impact of shipping in this region than in R13) and over the Southern Atlantic (10–30%, due to the higher background concentration of sulfate in this region than in R13). An interesting feature is the large contribution of shipping to the sulfate concentration over the Northern Atlantic Ocean, between the coasts of Greenland and Scandinavia: this is due to the increased emissions in the

polar regions in the CMIP6 dataset for 2015 compared to the CMIP5 for 2000, in particular along the Northern Sea Route, which has become more travelled in recent years following the depletion of the Arctic ice.

The shipping impacts are reduced in 2030 in all scenarios, although the geographical distribution remains essentially the same. This is because the SSPs do not consider any significant change in the route displacement, but just apply a scaling of the traffic activity across the existing routes according to a global spatial matrix (Feng et al., 2020). SSP1 and SSP2 (Fig. 5e,h)

project similar ship-induced concentrations, with the highest values reduced to 0.2–0.5 $\mu$g m$^{-3}$. In relative terms, however, this translates in a higher impact in SSP1, due to the cleaner background following the reductions in the emissions from other anthropogenic sectors. In SSP3 (Fig. 5k), the concentration changes in 2030 are still relatively high compared to 2015, although with a smaller relative contribution (Fig. 5l). The shipping sector has only a negligible impact on sulfate pollution in 2050 in SSP1 and SSP2 (Fig. 5n,o,q,r), while it remains relevant in SSP3, with large induced concentrations, around 0.2–0.5 $\mu$g m$^{-3}$

in the Northern Indian Ocean and around the Mediterranean, but with a relative contribution reduced to a maximum of 20% over the Northern Atlantic and Northern Pacific Ocean (Fig. 5t,u).

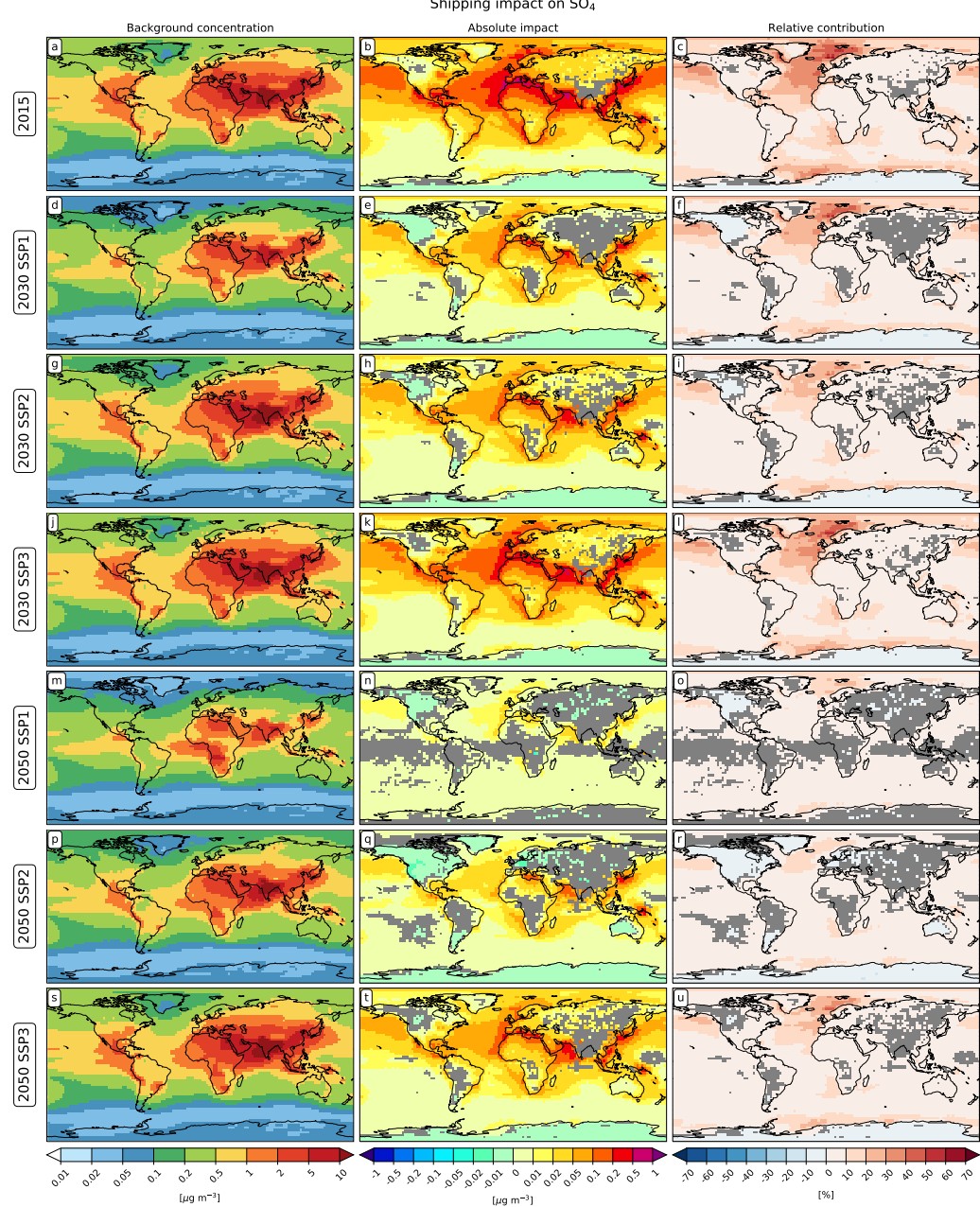

**Figure 5.** As in Fig. 3, but for the shipping impact on surface-level SO$_4$ concentration.

The impact on nitrate (Fig. S7) shows a similar geographic distribution, but a much different future evolution. The shipping-induced nitrate concentrations remain considerably high through 2030 and 2050, with very large relative contributions around and above 70% over most of the northern hemispheric oceans and along the major southern hemispheric routes. This reveals the

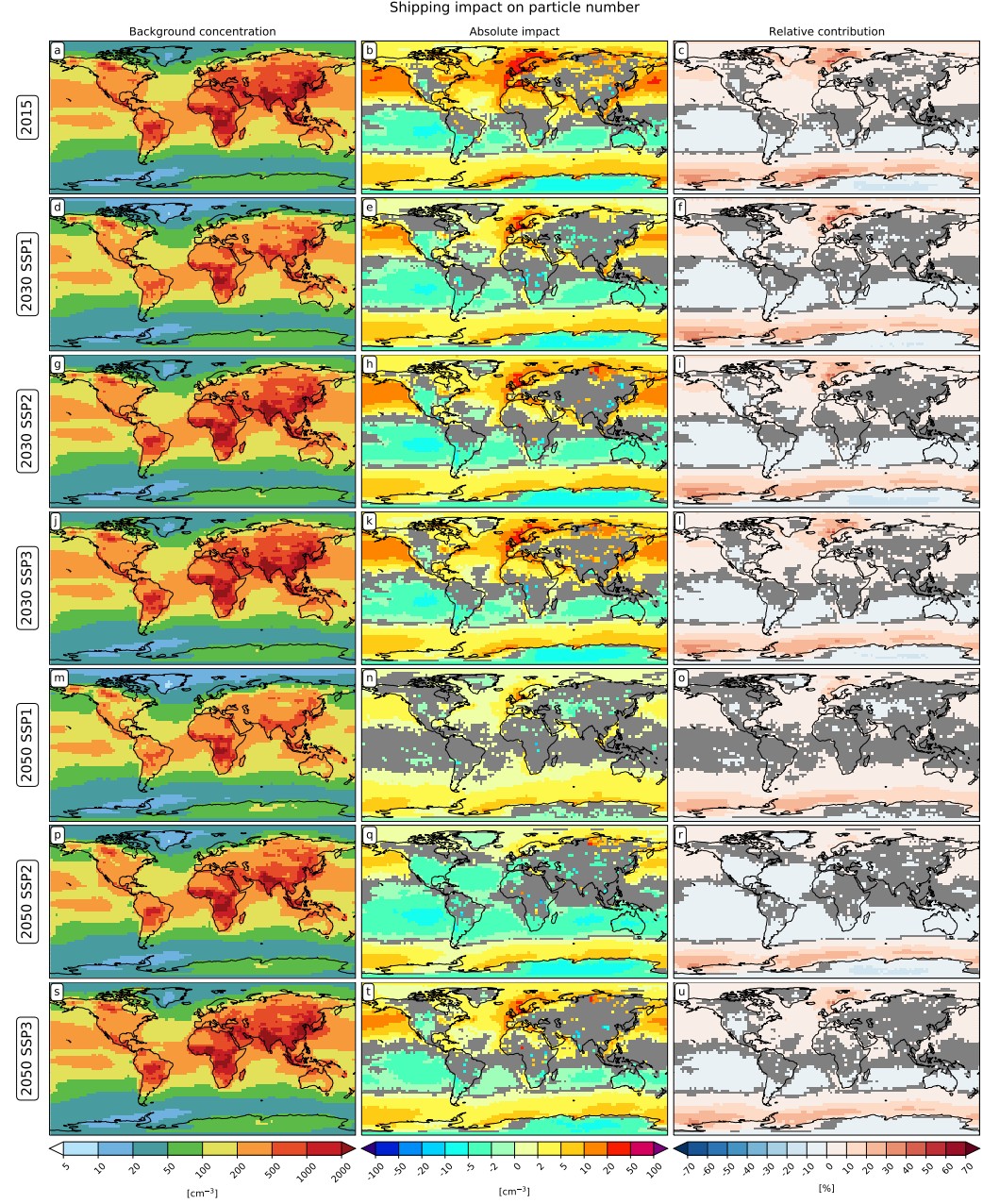

**Figure 6.** As in Fig. 3, but for the shipping impact on surface-level particle number concentration.

challenges in reducing the emissions from $NO_x$ in the shipping sectors, contrary to the sulfate which can be more effectively

controlled by reducing the shipping FSC. Nitrate impacts are effectively reduced only in the green SSP1 scenario in 2050.

Another aspect to be taken into account is the competition between the formation pathways of ammonium sulfate and ammo-





nium nitrate for available ammonia in the lower troposphere, with ammonium sulfate formation being favoured. A reduction of sulfate concentration makes more ammonia available for the formation of ammonium nitrate, leading to enhanced nitrate

formation, hence partly compensating the reduction in sulfate. Being the largest contributors to shipping pollution, sulfate and nitrate also influence the impact of this sector on particle number concentrations (Fig. 6). Here again the largest impacts are found over the northern hemispheric oceans and in the southern high latitude regions, progressively reducing in 2030 and 2050. Over the southern extra-tropics shipping induces a reduction of the particle number in most scenarios: this could be due to the relatively large size of ship-induced particles inhibiting the nucleation of new particles typical of the pristine environments of

these regions and thus limiting or removing a key source of particle number.

### 4.3 Aviation

Compared to the other two sectors, the impact of aviation on mass burdens is small (Fig. 2c), even though this is calculated in the altitude range of 8 to 12 km where most of the aviation emissions are released. Aviation contributes 0.3 Gg to the upper-tropospheric BC burden in 2015, almost three times more than in R13. The contribution is larger also in relative terms

(5% versus 2%). This could be due to an increase in BC emissions between 2000 and 2015, but also to the improved model vertical resolution adopted in the present study (41 levels) with respect to R13 (19 levels). The L41 resolution has been specifically designed to resolve the upper troposphere and the related process in higher detail and has been shown to be suitable in several modelling studies targeting this region (Dietmüller et al., 2014; Bock and Burkhardt, 2016; Righi et al., 2020). The coarse vertical resolution used in R13 may therefore have resulted in an underestimated impact of aviation-induced

BC in the upper troposphere. This appears to be the case also for sulfate, which shows an aviation-induced upper-tropospheric burden of 7.1 Gg, a factor of more than two larger than in R13. This is also larger in relative terms (4% vs. 3%). Despite the increased aviation impact on the mass burden, the impact on number is lower by about one order of magnitude than in R13 ($1.1 \times 10^{26}$ vs. $1.2 \times 10^{27}$ particles) and about the half in relative terms (6% vs. 12%), although identical size distributions for the emitted particles have been assumed in the two studies. Besides the aforementioned improvement in the vertical resolution,

the improved representation of the aerosol dynamics in the new version of the MADE aerosol submodel adopted here (MADE3; Kaiser et al., 2019) could explain these differences, as revealed, for instance, by the much lower background concentration of aerosol particles (Fig. 8a) with respect to the previous assessment. The EMAC model system has also been updated with respect to the previous assessment and now uses the second version of the MESSy interface, comprising numerous update in several key submodels, such as for example the scavenging submodel SCAV (Tost et al., 2006). Aviation-induced upper-tropospheric

BC burden keeps growing until 2030 in SSP2 and SSP3, and further in 2050 for SSP3. Aviation-induced BC burden decreases only in the SSP1 scenarios, with a factor of more than two reduction in 2030 and further down in 2050. In relative terms these changes are less marked, due to the corresponding changes in background BC concentrations. The sulfate burden follows a similar path, with a weak decrease in SSP1 and a steady increase in SSP2 and SSP3. This shows that the a desulphurisation of the aviation sector is not considered as an emission reduction measure in these scenarios, although it is relatively easy to

implement, as it does not require technical modifications nor does it need to cope with the long lifetime of the commercial fleet (e.g. Unger, 2011; Moore et al., 2017).





**Figure 7.** Aviation impact on zonally averaged BC concentration. The left panel show the background concentration for reference. The middle and right panels show the land-transport-induced absolute impacts and the relative contributions, respectively. Grid points where the differences are not significant at a 95% confidence level are masked out in gray. Note that, in contrast to the other figures, the units of the absolute changes are nanogram per cubic meter.

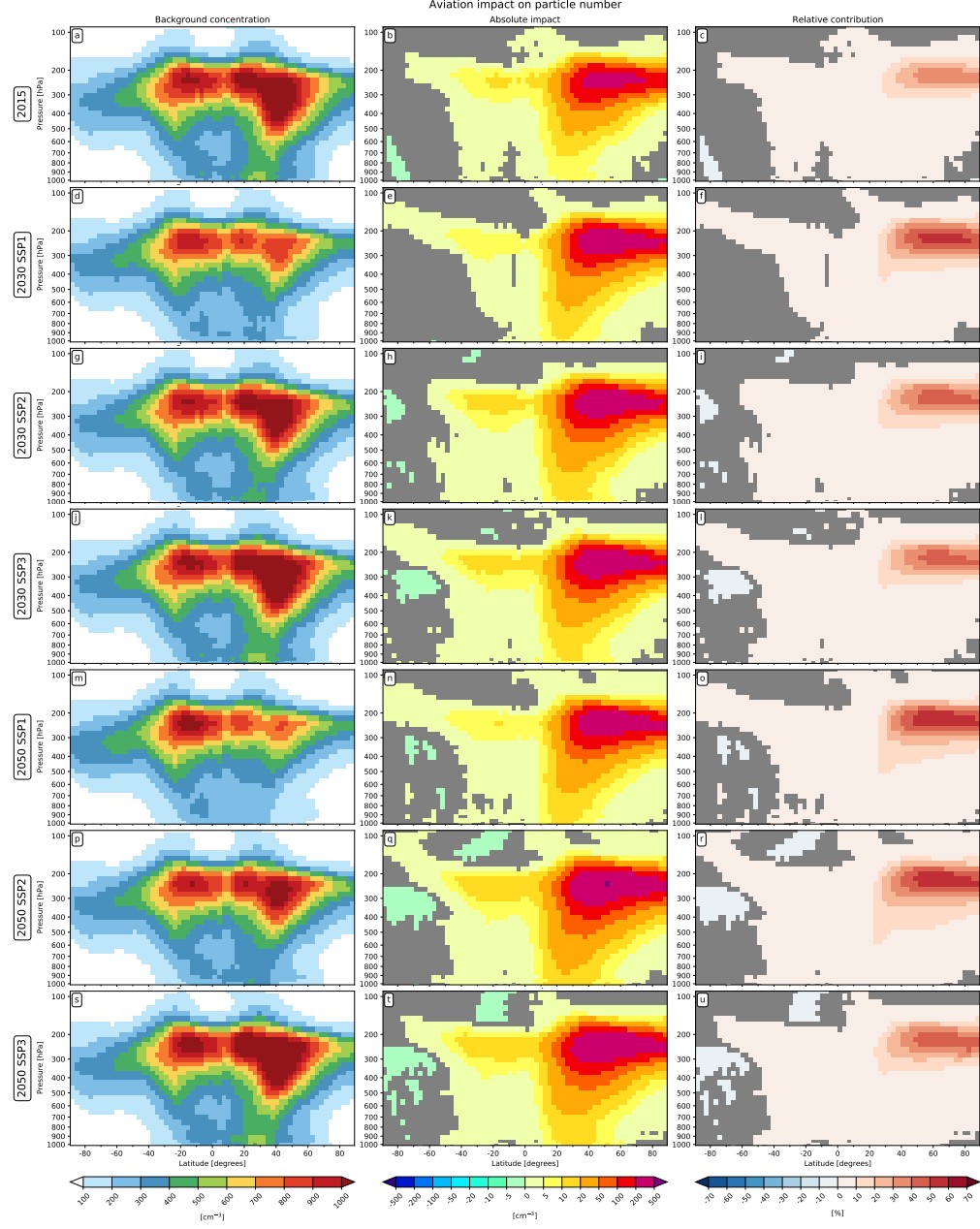

**Figure 8.** As in Fig. 7, but for the aviation impact on zonally averaged particle number concentration.

The spatial distributions of aviation-induced BC concentrations are depicted in Fig. 7 as zonal mean maps. In 2015 (Fig. 7b), the largest changes (0.5–1 ng m$^{-3}$) are simulated for the Northern Hemisphere at latitudes $> 40°N$ at the typical cruise levels of 200–350 hPa. Similarly high concentrations are simulated at lower levels, possibly driven by short- and medium-range



flight over the continents, and close to the surface at mid-latitudes, related to landing and take-off and airport activities. The Southern Hemisphere concentrations are only weakly affected by aviation, with induced changes in BC concentrations below 0.05 ng m$^{-3}$. As inferred from the analysis of the burdens, these values are significantly higher as those reported in R13. This is also the case for the relative contribution of aviation to BC concentration (Fig. 7c), peaking at 30–40% at high latitude in the Northern Hemisphere at 200–300 hPa. This is higher than the $\sim 5\%$ value reported in R13 and shifted northward, although the

overall pattern is still comparable. The future aviation impacts decrease only in the SSP1 scenario, while maintaining a similar zonal distribution. Aviation-induced BC concentrations are reduced to 0.2–0.5 ng m$^{-3}$ in 2030 and further down to 0.1 ng m$^{-3}$ over most of the northern latitudes in this scenario (Fig. 7e,n), with minor changes in relative terms (Fig. 7f,o). In SSP2, the absolute impact remains approximately constant until 2050 (Fig. 7h,q), while they increase in relative terms (Fig. 7i,l), up to 50–60% in 2050. This is the scenario with the highest relative contribution in the future: even though SSP3 shows much higher

absolute impacts (around 1–2 ng m$^{-3}$, Fig. 7k,t), the cleanest background emissions in SSP2 make the aviation sector more impacting in this scenario. Similar considerations can be made for the aviation-induced impacts on sulfate (Fig. S15), while the nitrate impacts (Fig. S13) show an interesting dual structure, with concentration increases between 5 and 20 ng m$^{-3}$ throughout the whole troposphere in northern mid-latitudes and significant increases in the upper troposphere at latitudes above 40°N. As mentioned in Sect. 4.2, this is due to the competition between ammonium sulfate and ammonium nitrate for available ammonia,

which is particularly distinct in the upper troposphere due to the low availability of ammonia (e.g. Unger et al., 2013).

Despite the relatively small impact on mass concentrations, of the order of a few nanograms per cubic meter, the aviation sector significantly increase the particle number concentration in the upper troposphere. This is due to the high efficiency of the combustion processes in aircraft engines, which lead to the formation of smaller BC particles compared to other sectors (Petzold et al., 1999). Furthermore, sulphur dioxide in aircraft plumes can drive efficient new particle formation in the upper

troposphere, contributing to particle number concentrations. Aviation-induced particle number concentrations as high as 200–500 cm$^{-3}$ are found in the upper-tropospheric northern mid-latitudes in 2015 (Fig. 8b). This is about one order of magnitude lower than in R13. The aviation impact also propagates to lower levels, with induced changes of a few tens of particles per cubic centimeter, with potential impacts on low level clouds. The strongest relative contribution of aviation to particle number concentration, however, remains confined in the upper troposphere at mid-latitudes, with values of 20–30%, in line with R13.

This means that both the aviation impact and the background are lower in this study than in R13. This could be due to the improved representation of the nucleation process in MADE3, which assumes a larger size of 10 nm for newly formed particles with respect to its predecessor version, in order to better reproduce the observations in the upper troposphere (Kaiser et al., 2019). As a consequence, the nucleation process leads to the formation of a lower number of particles (see again Fig. 8a).

The aviation impact on particle number concentrations remain large in all future scenarios. The impacts in the lower tropo-

sphere, in particular, increase to values larger than 10 cm$^{-3}$ in all SSPs, with potential consequences for the aerosol interactions with low clouds. Interestingly, the relative contribution show a counter-intuitive development, with the highest values in SSP1 and SSP2 ($> 60\%$) and smaller in SSP3. This can be explained with the to relatively long lifetime of the fleet in the aviation sector, leading to a slower implementation of technical and policy measures in this sector, which therefore reacts less promptly than the other sectors in determining the background concentrations. In SSP1, for example, background concentrations are very





efficiently reduced in 2030 and 2050 (Fig. 8d,m), while aviation follows a less efficient reduction path (Fig. 8e,n), resulting in a higher relative contribution to overall pollution in the future than in SSP3 (Fig. 8f,o). Note that, according to Feng et al. (2020), the geographical distribution of aviation emissions in the CMIP6 inventory remains the same as in 2015 in all SSPs and is simply scaled to match the global emissions.

## 5   Climate impacts

The climate impact of the transport sectors is depicted in terms of all-sky and clear-sky aerosol RF in Fig. 9. The first considers the changes in radiative fluxes on both cloudy and cloud-free model grid-boxes in each model time-step, whereas the latter is a diagnostic quantity calculated ignoring the effect of clouds on the radiative fluxes. The comparison of these two quantities helps understanding the role of aerosol-cloud interactions in aerosol-induced radiative forcing.

Aerosol from the land transport sector accounts for a RF of $-164$ mW m$^{-2}$ in 2015 (Fig. 9a), representing the largest
impact among the three sectors. This estimate is considerably larger than the value of $-12$ mW m$^{-2}$ reported in R13. Although both studies assume the same size distribution for the particles emitted by land transport (based on Birmili et al., 2009), here we further assume that primary sulfate aerosol particle from land transport are emitted in the Aitken mode with a geometric mean diameter of 30 nm (see Table 1). This results in a larger land-transport-induced impact on particle number concentration (Fig. 4b,c) than in R13, especially in relative terms, with land-transport contributions to particle number well above 20–30%
in several regions of the world. This is also due to a different representation of the aerosol particle background in the present study, which is generally lower than in R13 (Fig. 4a) and in better agreement with observations (Kaiser et al., 2019). The zonal profiles show that the effect is particularly strong in the northern mid-latitudes (Fig. S19a), where most emissions are released. The RFs simulated in the future scenarios clearly match the narratives underlying the SSPs. In SSP1, the land-transport RF is strongly reduced (in absolute terms) to $-28$ mW m$^{-2}$ in 2030 and becomes statistically non-significant in 2050. A less marked
reduction in the climate impact of land transport characterizes SSP2, with a RF of $-122$ mW m$^{-2}$ in 2030 and $-106$ mW m$^{-2}$ in 2050. Consistently with the narrative of the SSP3 scenario, an increase in RF is simulated here, with a stronger land-transport-induced cooling effect of $-194$ mW m$^{-2}$ in 2030 and $-205$ mW m$^{-2}$ in 2050. In all cases, the all-sky RF dominates over the clear-sky component, revealing that the land-transport-induced climate impact mostly results from the interactions between aerosol and clouds, thus confirming the results of R13 and Righi et al. (2015).

Aerosol from shipping exerts a RF of $-145$ mW m$^{-2}$ in the year 2015 (Fig. 9b), which is slightly smaller than the RF of $-181$ mW m$^{-2}$ found in R13. This is a consequence of the assumption on the size distribution of ship-emitted particles in this study: although we use the same size distribution parameters by Petzold et al. (2008), we assigned only 10% of the emitted mass to the Aitken mode rather than 20% as in R13, hence assuming a more aged population. This reduces the number of emitted particles and in turn the impact of these particles on clouds and the RF. Another reason, as discussed in
Sect. 4.2, is the lower shipping-induced aerosol sulfate in this study, following the introduction of the MARPOL regulations of FSC starting from 2010. Since sulfate has a remarkable contribution to cloud condensation nuclei, its reduction affects the RF impact via the aerosol indirect effect. As for land-transport, the shipping RF is significant at the northern mid-latitudes





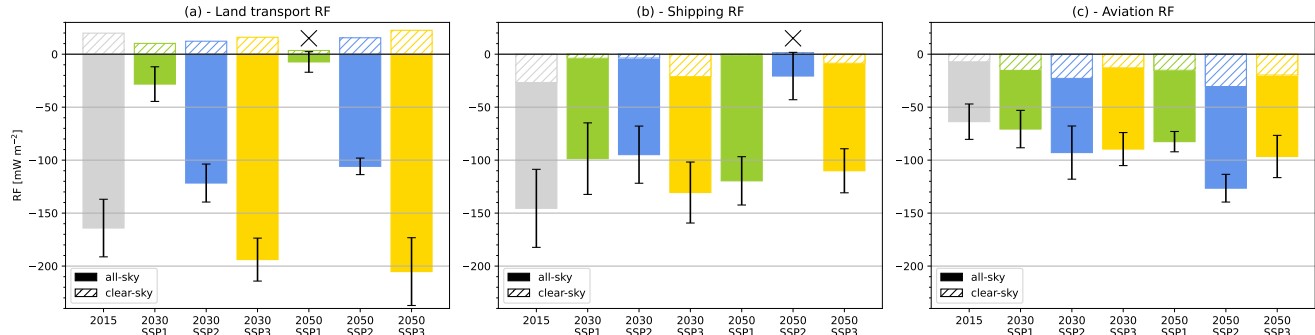

**Figure 9.** All-sky (solid) and clear-sky (hatched) RF of land transport, shipping and aviation in the year 2015 and for the three SSP scenarios in 2030 and 2050. The error bars represent the 95% confidence interval and refer to the all-sky RF. The crosses indicate non-significant results.

(Fig. S19b), but has an additional very strong component in the southern hemisphere, where ship traffic is still relevant along a few routes and, most importantly, aerosol and precursors are released in a pristine background (Fig. 6a), where the cloud susceptibility can be high. The shipping-induced negative RF is reduced in all scenarios in 2030, mostly for SSP1 and SSP2 (from $-145\,\mathrm{mW\,m^{-2}}$ to $-99$ and $-95\,\mathrm{mW\,m^{-2}}$, respectively), while in SSP3 the reduction is marginal (to $-131\,\mathrm{mW\,m^{-2}}$). In 2050, the greenest SSP1 scenario has the largest climate impact ($-120\,\mathrm{mW\,m^{-2}}$), comparable to SSP3 ($-110\,\mathrm{mW\,m^{-2}}$). This might appear counterintuitive, but can be explained with the much lower particle number background in this clean scenario (Fig. 6d), such that even a relatively small amount of shipping-induced particles leads to a large RF effect. SSP2, on the contrary, projects a non-significant aerosol climate effect from shipping in this year, consistent with the large reductions in shipping $SO_2$ emissions in this scenario (see Sect. 3 and Fig. 1d) and still relevant background sulfate pollution (Fig. 5p).

The impact of aviation-induced aerosol in 2015 is quantified as $-64\,\mathrm{mW\,m^{-2}}$, a factor of about four larger than the estimate in R13 ($-15\,\mathrm{mW\,m^{-2}}$). As discussed above, this is due to the larger relative impacts of aviation on particle number concentration in the current study, following an improved representation of background particles, in particular the ones generated by the new particle formation process (Kaiser et al., 2019). Although other modelling groups have confirmed these large aviation RF (Gettelman and Chen, 2013; Kapadia et al., 2016), the effect of aviation-induced aerosol on liquid clouds is still considered quite speculative (Lee et al., 2021), as large uncertainties exist in the size distribution of emitted particles and in the effectiveness of the downward transport of aviation-induced particles from typical flight altitudes (8–12 km) to the lower atmospheric layers, where they could potentially impact liquid phase clouds by acting as condensation nuclei. All SSPs project a steadily increasing impact of the aviation RF, with the strongest growth in SSP2, reaching a RF of $-126\,\mathrm{mW\,m^{-2}}$ in 2050. As argued in Righi et al. (2016), the long lifetime of the aviation fleet and the tight constraints on safety and costs make the introduction of mitigation measures in aviation more challenging than in other sectors. In all scenarios the mitigation measures do not lead to a reduction of the overall aviation emissions, which are instead only modulated by different growth rates in overall traffic volumes.



# 6 Conclusions and outlook

In this study, the global impact of the transport sectors (land transport, shipping and aviation) on aerosol and the resulting climate effects for present-day (2015) and three future projections for 2030 and 2050 have been assessed using the most recent CMIP6 emission inventory and the SSPs scenarios as an input to a state-of-the-art global aerosol-climate model, including detailed aerosol microphysics and an explicit representation of aerosol-radiation and aerosol-cloud interactions. The results have been compared with the findings of our previous assessments (Righi et al., 2013, 2015, 2016), which used a predecessor version of the same model and the earlier CMIP5 emissions data for the year 2000 combined with the RCP projections for 2030.

Black carbon and nitrate are found to be highly relevant pollutants from land transport, which contributes 15–20% to the near-surface concentrations of these species. The shipping sector is also an important source of nitrate pollution, responsible for more than 20% of its near-surface concentration over the oceans, and still contributes significantly to aerosol sulfate pollution, despite the introduction of control measures for fuel sulphur content in shipping by MARPOL since 2010. The aviation contribution to the aerosol mass concentrations in the upper-troposphere is generally small (below 5%), while it is larger for particle number (5–10%). The results for the transport-induced aerosol mass concentrations generally confirm the findings of R13, although some major differences exist, especially in the geographical distribution of the transport impacts. These are essentially due to the updated emission inventory used as an input to the model simulations and the different year under investigation (2015 vs. 2000), but also to the improvement of the model. Substantial differences, on the other hand, are found in terms of simulated transport-induced impacts on aerosol number concentration. This is because the new MADE3 model version adopted here is characterized by a generally lower background concentration of particles, as a result of different assumptions on the particles size distribution upon emissions by anthropogenic sources (including transport) and an improved representation of the aging of newly-formed particles via the nucleation process. The extensive evaluation performed by Kaiser et al. (2019) and Righi et al. (2020) shows that the new model version provides a better representation of the aerosol background and the climate mean-state of the key cloud and radiation variables compared with observations. This results in larger RF effects for land transport and aviation with respect to the previous assessment, while a smaller RF is simulated for shipping, thanks also to the reduced sulfate impacts following the MARPOL regulations on fuel sulphur content.

The mid- and long-term projections of the transport effects under the SSPs are generally consistent with the narratives underlying the scenarios. SSP1 is characterized by the lowest impacts, with a strong decrease of the land transport effects, a slight decrease for shipping and a lower increase than the other scenarios for aviation. The climate impact of shipping and aviation in this scenario are still relatively high despite a strong reduction in their emissions, which can be ascribed to the very low background concentrations, which is mostly controlled by the emissions of the other anthropogenic sectors. The results for the SSP2 scenario confirm its middle-of-the-road nature, which is evident especially in the impacts of the land transport sector. A peculiarity of this scenario it the low climate impact of shipping, due to low emissions from this sector combined with still elevated aerosols levels induced by other sectors, and the relatively large impact of aviation, whose emissions in this scenario are still high compared to other sectors. The strong growth of pollution characterizing SSP3 is remarkable, showing steadily



growing impacts in 2030 and 2050 with respect to 2015, especially in the land transport sector. These results show that the
impact of the transport sectors under the SSPs are more consistent with the respective storylines than in the predecessor RCP
scenarios used in (Righi et al., 2015, 2016).

From the analysis of the simulations in this study we can conclude that the challenges in quantifying the impact of the trans-
portation sectors on aerosol and climate are twofold. On the one hand, reliable emission inventories are essential for a correct
quantification of the resulting impacts: this concerns not only the total emitted mass, but also its geographical distribution. On
the other hand, the representation of the aerosol concentration background in the model can play a significant role in charac-
terizing the impact of the perturbation introduced by the emissions of the specific sectors. This is particularly the case when
analysing the scenario projections, as the potential reductions in the emissions for a given sector needs to be set in the context of
the emission changes in other sectors. Policy measures cannot therefore be evaluated only focusing on a single sector, but have
to be considered as part of a larger picture. The representation of the background is also a modelling challenge and our results
suggest that an accurate representation of the aerosol background concentrations is important and can significantly affect the
estimated impacts, as it is clear when comparing the outcome of the present study with the result of the previous assessment
obtained with a predecessor version of the same model. The prominent role of the background has further consequences for
the development and application of climate response models (e.g., Grewe and Stenke, 2008; Wild et al., 2012; Dahlmann et al.,
2016; Rieger and Grewe, 2022) to assess the transportation impacts of aerosol, implying that the global model simulations for
training these models shall be performed of varying background conditions on the region of interest.

On the policy side, a further challenge is posed by the trade-offs connected with the implementation of specific measures. One
prominent example is the reduction in fuel sulfur content in shipping to improve air quality in the coastlines, that at the same
time leads to a reduction of the ship-induced cooling via the aerosol indirect-effect, in turn reducing the offset of greenhouse
warning (Quaas et al., 2022). Future modelling studies will also need to deal with the growing impacts of alternative fuels and
their consequence on the atmospheric composition, for example hydrogen (Popa et al., 2015), and new particle types resulting
from non-combustion processes (Grigoratos and Martini, 2014; Timmers and Achten, 2016), whose impact is projected to
become increasingly important with the progressive electrification of the land transport sector.

*Code and data availability.* MESSy is continuously developed and applied by a consortium of institutions. The usage of MESSy and access
to the source code is licensed to all affiliates of institutions which are members of the MESSy Consortium. Institutions can become members
of the MESSy Consortium by signing the MESSy Memorandum of Understanding. More information can be found on the MESSy Consor-
tium Website (http://www.messy-interface.org, last access: 22 November 2022). The model configuration discussed in this paper is based on
EMAC version 2.55. The exact setup used to produce the result of this paper is archived at the German Climate Computing Center (DKRZ)
and can be made available to members of the MESSy community upon request. The output of the model simulations discussed in this work
will be made available via DOI in the final version of this paper.



*Author contributions.*  MR conceived the study, designed and performed the simulations, analysed the data and wrote the paper. JH conceived the study, contributed to the simulation design and to the text. SB prepared the CMIP6/SSP emission data used to drive the model simulations and contributed to the text.

*Competing interests.*  The authors declare no competing interests.

*Acknowledgements.*  We are grateful to Axel Lauer (DLR) for his valuable comments on an earlier version of this manuscript, and to Christof
Beer, Patrick Jöckel and Mariano Mertens (DLR) for helpful suggestions. The processing of the CMIP6/SSP emissions for usage in EMAC has greatly benefited from the efforts by Markus Kunze (Freie Universität Berlin) and Phoebe Graf (DLR). MR would also like to thank Manuel Schlund (DLR), for his competent help with the matplotlib Python library. The model simulations and data analysis for this work used the resources of the Deutsches Klimarechenzentrum (DKRZ) granted by its Scientific Steering Committee (WLA) under project ID bd0080.

*Financial support.*  This study was supported by the DLR transport research program (TraK and DATAMOST projects).



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
