# Peer review of "The global impact of the transport sectors on the atmospheric aerosol and the resulting climate effects under the Shared Socioeconomic Pathways (SSPs)"

_Earth System Dynamics, 2022_

## Author Comment (AC1)

***The global impact of the transport sectors on aerosol and climate under the***
***Shared Socioeconomic Pathways (SSPs)***
**Mattia Righi, Johannes Hendricks and Sabine Brinkop**
**Replies to referees' comments**

We are grateful to the two reviewers for their insightful comments and constructive criticism, which greatly helped us to improve the manuscript.

A detailed reply to each comment can be found below (reviewers' comments are marked in blue, authors' reply in black, and text quotes in *"italic red"*).

**Anonymous referee #1**

Regarding the title: "The global impact of the transport sectors on aerosol and climate under the Shared Socioeconomic Pathways (SSPs)", it can be interpreted as the work consider the global impact of transport on climate, but it is only the aerosol impact of the transport sector on climate that is considered. Consider rephrasing, also in the abstract.

We have changed the title as suggested by Reviewer #2, who raised a similar concern. It now reads: *"The global impact of the transport sectors on the atmospheric aerosol and the resulting climate effects under the Shared Socioeconomic Pathways (SSPs)"*

Regarding the above, is it possible to add a discussion regarding the CO2 warming from the sectors in the scenarios?

The role of the other forcers will be assessed in two companion manuscripts which are currently in preparation, focusing respectively on ozone impacts and on the overall impacts of the transport sectors, comparing $CO_2$ and non-$CO_2$ effects.

We have added a sentence at the end of the manuscript to make the readers aware of this: *"This paper focused on the impact of the transport sectors on aerosol and the related climate effects in the SSPs. Two companion studies are currently ongoing to address the transport impact on ozone and the overall impacts of the transport sectors (comparing $CO_2$ and non-$CO_2$ effects), respectively."*

In the abstract: "In spite of the recently introduced regulations to limit the fuel sulphur content in the shipping sector, shipping emissions are still responsible for a considerable impact on aerosol sulfate near-surface concentrations", I wonder how well the scenarios take into account the most recent regulations.

Unfortunately, the respective literature (see references in the manuscript) is not very exhaustive on this issue. A quick analysis of the geographical distribution of $SO_2$ emission ratios 2030/2015 and 2050/2015 shows, however, that different assumptions were done for the SECA regions and the US coastal waters compared to the open oceans. This suggests that the impact of the regulations has been (at least partly) considered by the scenarios. But it is challenging to sort out which specific assumption (fleet composition, traffic volumes, fuel types, routes displacement, etc.) is responsible for the emission changes.

Although this has improved over the previous generation of scenarios (RCPs), the SSP documentation still lacks important information which would help to better interpret the results for specific sectors.

L39: "some" I will say "all", as all of these components will either directly or indirectly (via chemistry) affect climate.

Good point, we have rephrased the sentence: *"These compounds have a detrimental effect on the air quality and can also impact climate"*.

L57: CMIP6 emission inventory (Hoesly et al., 2018; van Marle et al., 2017) Hoesly et al. is the historical anthropogenic emissions used in CMIP6 that end in 2014.

Correct. We have replaced the two references with Gidden et al. (2019) and Feng et al. (2020).

L171: Related to above: "and future projections until the year 2100" the two references cover the historical period and not the future projections.

Same as above, we have replaced them with Gidden et al. (2019) and Feng et al. (2020).

I find the two first paragraphs in section 3 on the emissions a bit unstructured. When reading, I was expecting the matrix structure of the SSPs to appear earlier. The link and harmonization of the historical and the scenarios could be made clearer, as described in Gidden et al. 2019. 2015 emissions in the harmonized emissions are similar in all the scenarios.

The description of the scenarios was indeed a bit confusing. We have restructured the first part of Sect. 3, which should be clearer now.

There are a wide range of SSP scenarios, but for climate modelling, only a small selection is gridded and harmonized for use. For SSP1 and SSP3, there are two other forcing targets available. Would be good if the differences related to aerosol emissions in these could be slightly mentioned.

We agree with this suggestion and have added a paragraph in Sect. 3: *"As mentioned above, various combinations of the SSPs with the forcing pathways until 2100 are available. For example, for SSP1 the SSP1-2.6 pathway is also available in addition to the SSP1-1.9 adopted here. These two pathways show a similar decreasing emission trend in the future, but with some differences in their total emissions: for $NO_x$, $SO_2$ and BC, for instance, the emissions in 2050 are about 52%, 24% and 29% larger in the SSP1-2.6 pathway. Similarly, for SSP3 a so-called LowNTCF pathway is also available, characterized by a strong decrease in the emissions of near-term climate forcers until 2050, followed by a slowly decreasing trend. With respect to the SSP3-7.0 case analysed in this study, the total emissions of $NO_x$, $SO_2$ and BC in 2050 are therefore significantly lower in this pathway, namely about 51%, 46% and 54%, respectively. Choosing a different pathway within a given SSP has of course an impact on the results. Nevertheless, the three pathways chosen for this assessment reasonably cover the range of possible future emission trends until 2050."*

Figure 2: Add to the figure caption the domain over which the burden is calculated.

Good point, it has been added.

All figures: Replace the rainbow color scale with a different color scale.

Thanks for this suggestion. We have replaced the rainbow color scale with a sequential one in all relevant figures (also in the supplement).

5 Climate impacts. In the beginning of this section, it would be useful with a clearer definition of the radiative forcing definition. Do you calculate RF or ERF (including the adjustments). Is it both radiative forcing for aerosol cloud interaction (aci) and aerosol radiation interaction (ari)? And more specifically, how to compare the clear sky and all sky RF to separate the role of aci and ari.

To address these questions, we have rephrased and extended the first paragraph of Sect. 5, which now reads as follows:

*"To quantify the climate impacts we apply the well-known RF metric (Ramaswamy et al., 2019). More specifically, the climate impact of the transport sectors is estimated as the difference in the radiative fluxes between a reference and a sensitivity simulation, the latter completely neglecting the emissions of the given sector (see also Sect. 2). Given that we are using an aerosol-cloud coupled model, the radiative fluxes and the resulting forcings calculated here explicitly consider the impact of cloud adjustments, hence the RF values presented in the following can be regarded as effective radiative forcings. Furthermore, the model is able to diagnose both the all-sky and the clear-sky aerosol RF: The first considers the changes in radiative fluxes on both cloudy and cloud-free model grid-boxes in each model time-step, whereas the latter is a diagnostic quantity*

*calculated ignoring the effect of clouds on the radiative fluxes. The difference between these two quantities can be regarded as a proxy for the RF due to aerosol-cloud interactions. Note, however, that the clear-sky RF cannot be ascribed to the aerosol-radiation interactions only, as it may include other forcings too (e.g, the ones due changes in water vapour or to the semi-direct effect). These caveats should be kept in mind when interpreting the results discussed in this section."*

Figure 9: How is the 90% CI calculated?
As mention at the end of Sect. 2, the confidence interval is calculated using two-tailed t-test with respect to the interannual variability. We have added this information in the caption of Fig. 9 for more clarity.

L533-549: This section conclude the results for present day 2015? Indicate that in the first sentence.
Good point, we have added *"In the simulations for present-day (2015) conditions"* at the beginning of this paragraph.

L580: Any studies on ammonia in the transport sector?
We have extended the sentence and added two references on ammonia and also methanol, which could be relevant for the shipping sector. Thank you for suggesting this!

**Anonymous Referee #2**
The scope of the study is the assessment of the global transportation sector on the atmospheric aerosol burden and the resulting effects on the climate. The title, however, suggests a slightly different topic. A more specific title could be: "The global impact of the transport sectors on the atmospheric aerosol and resulting climate effects under the Shared Socioeconomic Pathways (SSPs)".
Thank you for your suggestion: we have changed the title accordingly, also to address a similar concern by Reviewer #1.

In Section 3, the authors discuss the emission of aerosol and precursors for today's conditions and future scenarios. Emission scenarios are discussed on the basis of changing activities by scaling the aerosol properties of current sources. However, the aerosol characteristics may change significantly with respect to chemical composition and resulting changes in aerosol-cloud-interactions when new fuels are used; see, e.g., Petzold et al. (2005) and Moore et al. (2017) for aviation and Petzold et al. (2011) for shipping. These studies report changing aerosol properties when switching to low-sulphur fossil or biogenic fuels. The potential impact of those changing aerosol properties should be mentioned and discussed.
We have added a concluding statement in Sect. 3 to discuss this limitation: *"The assumptions on the particle size distributions summarized in Table 1 are based on measurements performed in past years and are not necessarily representative of the future conditions. Under the SSP scenarios changes can be expected not only in terms of emission amounts, as discussed above, but also in terms of properties of the emitted particles, such as size distribution, chemical composition and mixing state. A few studies, for example, show how these properties change when different fuel types are used in aviation (Petzold et al., 2005; Moore et al., 2017) and in shipping (Petzold et al., 2011). This might have an impact not only on the size and number concentration of the emitted particles, but also on their chemical properties and mixing states, thus affecting their ability to act as cloud condensation nuclei and their resulting climate impacts. Ideally, different assumptions should be made for each scenario, depending, for instance, on the properties of the fleet and the fuel types implemented in them. This is an aspect that could be considered in the next generation of scenarios, to better support modelling studies and assessments as the one presented in this work."*

This comment refers also to the statement on line 91ff, "that particle number emission factors in low-sulphur fuels are not significantly decreased compared to standard heavy fuel oil". Although the total number

concentrations may not change significantly, the chemical nature of the particles changes and thus the interaction with clouds.
The sentence has been removed since it was indeed misleading.

In Section 4, the Transport impact on aerosol concentrations is discussed. The section presents the results very clearly, but the potential impact of changing background conditions should at least be mentioned. As an example, how will the relative contribution of the transport sector to the BC burden look like when the total atmospheric BC load changes under heavy forest fire conditions? A short discussion might give more insight on the relevance of the presented results in a changing environment.
We agree that the role of a changing background is important when interpreting the relative impacts of the transport sectors on aerosol concentrations in the future scenarios. This is indeed already addressed in the manuscript for a few prominent cases in of the shipping (L406-408) and the aviation (L447-448, L465-L467, L490-492) sectors. This is further highlighted in the discussion on climate impacts, to explain the (apparently counterintuitive) larger impact of shipping in SSP1 than in SSP3 (L535-539). In the conclusion, we summarized the importance of the background as follows: *"On the other hand, the representation of the aerosol concentration background in the model can play a significant role in characterizing the impact of the perturbation introduced by the emissions of the specific sectors. This is particularly the case when analysing the scenario projections, as the potential reductions in the emissions for a given sector needs to be set in the context of the emission changes in other sectors. Policy measures cannot therefore be evaluated only focusing on a single sector, but have to be considered as part of a larger picture. The representation of the background is also a modelling challenge and our results suggest that an accurate representation of the aerosol background concentrations is important and can significantly affect the estimated impacts, as it is clear when comparing the outcome of the present study with the result of the previous assessment obtained with a predecessor version of the same model. The prominent role of the background has further consequences for the development and application of climate response models (e.g., Grewe and Stenke, 2008; Wild et al., 2012; Dahlmann et al., 2016; Rieger and Grewe, 2022) to assess the transportation impacts of aerosol, implying that the global model simulations for training these models shall be performed of varying background conditions on the region of interest."*
We hope this addresses your question.

Line 78: Please correct to "… implementations of reduction measures".
Done.

Line 81: Which fleet is referred to here? Please clarify.
We refer here to the land transport fleet. It has been added.

Line 149: The authors use ERA-Interim reanalysis data. Would the use of ERA 5 reanalysis data produce significant differences?
As part of another study, we tested the impact of using different reanalysis data on our model's results and found this to be mostly non-significant for the typical climate variables (temperature, winds, humidity, geopotential, etc…).

Figure 2: The y-axis title of the first row is confusing and might be understood as mass normalized to particle number. Please check.
We have replaced the "/" symbol with "or" to avoid confusion.

Line 556: Please correct to "A peculiarity of this scenario is the low climate impact …"
Fixed, thanks for spotting this and the above typos.